# Associations between Clinical Symptoms and Degree of Ossification in Patients with Cervical Ossification of the Posterior Longitudinal Ligament: A Prospective Multi-Institutional Cross-Sectional Study

**DOI:** 10.3390/jcm9124055

**Published:** 2020-12-15

**Authors:** Takashi Hirai, Toshitaka Yoshii, Shuta Ushio, Jun Hashimoto, Kanji Mori, Satoshi Maki, Keiichi Katsumi, Narihito Nagoshi, Kazuhiro Takeuchi, Takeo Furuya, Kei Watanabe, Norihiro Nishida, Soraya Nishimura, Kota Watanabe, Takashi Kaito, Satoshi Kato, Katsuya Nagashima, Masao Koda, Kenyu Ito, Shiro Imagama, Yuji Matsuoka, Kanichiro Wada, Atsushi Kimura, Tetsuro Ohba, Hiroyuki Katoh, Masahiko Watanabe, Yukihiro Matsuyama, Hiroshi Ozawa, Hirotaka Haro, Katsushi Takeshita, Morio Matsumoto, Masaya Nakamura, Masashi Yamazaki, Masato Yuasa, Hiroyuki Inose, Atsushi Okawa, Yoshiharu Kawaguchi

**Affiliations:** 1Department of Orthopedic Surgery, Tokyo Medical and Dental University, Tokyo 113-8510, Japan; yoshii.orth@tmd.ac.jp (T.Y.); ushiorth20@gmail.com (S.U.); 0123456789jun@gmail.com (J.H.); yuasa.orth@tmd.ac.jp (M.Y.); inose.orth@tmd.ac.jp (H.I.); okawa.orth@tmd.ac.jp (A.O.); 2Japanese Organization of the Study for Ossification of Spinal Ligament (JOSL), Tokyo 113-8510, Japan; kanchi@belle.shiga-med.ac.jp (K.M.); satoshi.maki@chiba-u.jp (S.M.); kkatsu_os@yahoo.co.jp (K.K.); nagoshiation@gmail.com (N.N.); takeuchi@okayamamc.jp (K.T.); takeo251274@yahoo.co.jp (T.F.); keiwatanabe_39jp@live.jp (K.W.); nishida3@yamaguchi-u.ac.jp (N.N.); soraya.nishimura@gmail.com (S.N.); watakota@gmail.com (K.W.); takashikaito@gmail.com (T.K.); skato323@gmail.com (S.K.); katsu_n103@yahoo.co.jp (K.N.); masaokod@gmail.com (M.K.); kenyubankara@yahoo.co.jp (K.I.); imagama@med.nagoya-u.ac.jp (S.I.); yuji_kazu77@yahoo.co.jp (Y.M.); wadak39@hirosaki-u.ac.jp (K.W.); akimura@jichi.ac.jp (A.K.); tooba@yamanashi.ac.jp (T.O.); hero@tokai-u.jp (H.K.); masahiko@is.icc.u-tokai.ac.jp (M.W.); spine-yu@hama-med.ac.jp (Y.M.); hozawa@med.tohoku.ac.jp (H.O.); haro@yamanashi.ac.jp (H.H.); dtstake@gmail.com (K.T.); morio@a5.keio.jp (M.M.); masa@a8.keio.jp (M.N.); masashiy@md.tsukuba.ac.jp (M.Y.); zenji@med.u-toyama.ac.jp (Y.K.); 3Department of Orthopaedic Surgery, Shiga University of Medical Science, Shiga 520-2192, Japan; 4Department of Orthopedic Surgery, Chiba University Graduate School of Medicine, Chiba 260-0856, Japan; 5Department of Orthopedic Surgery, Niigata University Medical and Dental General Hospital, Niigata 951-8520, Japan; 6Department of Orthopedic Surgery, Keio University School of Medicine, Tokyo 160-8582, Japan; 7Department of Orthopedic Surgery, National Hospital Organization Okayama Medical Center, Okayama 701-1192, Japan; 8Department of Orthopedic Surgery, Yamaguchi University Graduate School of Medicine, Yamaguchi 755-8505, Japan; 9Department of Orthopaedic Surgery, Osaka University Graduate School of Medicine, Osaka 565-0871, Japan; 10Department of Orthopedic Surgery, Graduate School of Medical Sciences, Kanazawa University, Ishikawa 920-1192, Japan; 11Department of Orthopedic Surgery, Faculty of Medicine, University of Tsukuba, Ibaraki 305-8577, Japan; 12Department of Orthopedic Surgery, Nagoya University Graduate School of Medicine, Aichi 464-8601, Japan; 13Department of Orthopedic Surgery, Tokyo Medical University, Tokyo 160-8402, Japan; 14Department of Orthopedic Surgery, Hirosaki University Graduate School of Medicine, Aomori 036-8562, Japan; 15Department of Orthopedics, Jichi Medical University, Tochigi 329-0498, Japan; 16Department of Orthopedic Surgery, University of Yamanashi, Yamanashi 400-8510, Japan; 17Department of Orthopedic Surgery, Surgical Science, Tokai University School of Medicine, Kanagawa 259-1193, Japan; 18Department of Orthopedic Surgery, Hamamatsu University School of Medicine, Shizuoka 431-3125, Japan; 19Department of Orthopaedic Surgery, Tohoku Medical and Pharmaceutical University, Miyagi 981-8558, Japan; 20Department of Orthopedic Surgery, Faculty of Medicine, University of Toyama, Toyama 930-8555, Japan

**Keywords:** ossification of the posterior longitudinal ligament (OPPL), pain, OP-index, canal narrowing ratio, spinal disorder, patient-reported outcomes, computed tomography, activities of daily living, quality of life

## Abstract

This study aimed to clarify whether ossification predisposition influences clinical symptoms including pain, restriction of activities of daily living, and quality of life in patients with cervical ossification of the posterior longitudinal ligament (OPLL). Cervical ossification predisposition potentially causes neurologic dysfunction, but the relationship between clinical symptoms and radiologic severity of OPLL has not yet been investigated. Data were prospectively collected from 16 institutions across Japan. We enrolled 239 patients with cervical OPLL. The primary outcomes were patient-reported outcomes, including visual analog scale (VAS) pain scores and other questionnaires. Whole-spine computed tomography images were obtained, and correlations were investigated between clinical symptoms and radiologic findings, including the distribution of OPLL, the sum of the levels where OPLL was present (OP-index), and the canal narrowing ratio (CNR) grade. The cervical OP-index was Grade 1 in 113 patients, Grade 2 in 90, and Grade 3 in 36. No significant correlations were found between radiologic outcomes and VAS pain scores. The cervical OP-index was associated with lower extremity function, social dysfunction, and locomotive function. The CNR grade was not correlated with clinical symptoms, but Grade 4 was associated with lower extremity dysfunction. Thickness and extension of ossified lesions may be associated with lower extremity dysfunction in cervical OPLL.

## 1. Introduction

Ossification of spinal ligaments is recognized as a musculoskeletal disorder that often leads to neurological dysfunction. Ossification of the posterior longitudinal ligament (OPLL) has been widely investigated worldwide because of its high prevalence. This heterotopic ossification occurs particularly in the cervical and thoracic spine, where the spinal cord lies inside the canal space, and can result in spinal disorders, even in asymptomatic cases following minor head trauma. Generally, in patients with OPLL, once the onset of myelopathic symptoms has started, neurologic dysfunction gradually deteriorates, and the patient will then require decompressive surgery [1]. Therefore, it is important that patients and physicians alike recognize the presence and location of OPLL and whether the entire spine is involved.

Various studies on OPLL have described numerous issues, such as the efficacy of surgical treatments [2,3,4,5,6,7], clinical courses [8,9], and epidemiologic data [10,11]. Recent studies have demonstrated multiple ossified lesions in patients with ossification predisposition [12,13,14,15,16,17,18]. A retrospective multicenter study [12] showed that more than 55% of patients with cervical OPLL also had OPLL at the thoracic and/or lumbar spine. In addition, OPLL has been shown to coexist with the ossification of other spinal ligaments, including the nuchal ligament [13], the anterior longitudinal ligament [14], the supra- and inter-spinous ligaments [15], the yellow ligament in the whole spine [16]. Ossification of these ligaments, especially in the cervical spine, sometimes leads to neurologic symptoms, including numbness or loss of sensation in the extremities and difficulty walking. Expansion of ossified lesions in the whole spine is represented by the OPLL ossification index (OP-index) [17,18], which is the sum of the vertebral body and intervertebral disc levels where OPLL is present. This index has a significantly positive association with female sex, obesity, and the extent of cervical OPLL [12]. Although it is clinically known that cervical ossification predisposition correlates to the potential for neurologic dysfunction, no study has investigated the relationship between clinical characteristics and radiologic severity of ossified lesions in OPLL patients. Therefore, we conducted a prospective multicenter investigation to clarify whether ossification predisposition can influence clinical symptoms including body pain, restriction of activities of daily living (ADL), and quality of life in patients with cervical OPLL. This study focused on the spatial expansion of OPLL assessed using the cervical OP-index and the thickness of ossification based on the canal narrowing ratio (CNR).

## 2. Materials and Methods

### 2.1. Patients and Methods

This multicenter prospective cross-sectional study included 16 institutions of the Japanese Multicenter Research Organization for Ossification of the Spinal Ligament established by the Japan Ministry of Health, Labour and Welfare. To be eligible for the study, patients had to meet all the following criteria: ≥20 years of age; a cervical OPLL diagnosis based on X-ray findings; symptoms such as neck pain, upper and/or lower extremity numbness (regardless of whether surgery was required), clumsiness, and gait disturbance; had visited a participating hospital for symptoms between September 2015 and December 2017; and had undergone whole-spine computed tomography (CT) scan to determine the location of ossified lesions in the whole spine. We excluded patients who had undergone anterior decompression surgery or posterior OPLL surgery.

The study protocol was approved by the institutional review board of each participating institution and the study was conducted in accordance with relevant guidelines and regulations. Informed consent was obtained from all patients before enrollment in the study.

### 2.2. Clinical Evaluations

Basic demographic and clinical data, including age and sex, diabetes mellitus (DM), body mass index (BMI), and the presence of neck pain, back pain, and low back pain (LBP), were collected for each patient (Appendix A). Clinical status was evaluated using the cervical Japanese Orthopedic Association (JOA) score [19], the JOA Cervical Myelopathy Evaluation Questionnaire (JOA-CMEQ) [20], and the JOA Back Pain Evaluation Questionnaire (JOA-BPEQ) [20]. A visual analog scale (VAS) was used to evaluate the degree of pain or stiffness in the neck or shoulders, pain or numbness in the arms or hands, and LBP.

### 2.3. Radiologic Evaluations

CT images of the whole spine—including the cervical, thoracic, and lumbosacral spine from the occipital bone to the sacrum—were obtained for each patient. OPLL incidence in the cervical spine, from the clivus to C7 and in other spinal regions from T1 to S1, was evaluated on mid-sagittal CT images. The analysis was performed independently by 6 senior spine surgeons (S.U., K.M., S.M., K.K., N.N., and K.T.) who were blinded to clinical outcomes. Before the review, all testers read the same images of 20 patients to check the inter-observer agreement. The mean Kappa coefficient of the inter-observer agreement was 0.83 (0.79–0.85), indicating substantial agreement and consistency with previous results [12]. Observed ossified lesions were recorded for each vertebral body and intervertebral disc level (Appendix A). The OP-index was calculated as described previously [12,17]. The number of OPLL lesions in the cervical spine was designated by the cervical OP-index. As previously validated [12], we categorized patients into three groups by cervical OP-index: Grade 1, cervical OP-index ≤5; Grade 2, cervical OP-index 6–9; and Grade 3, cervical OP-index ≥10 (Figure 1). In addition to the OP-index, the sum of the intervertebral segments where each ossification was located, including the anterior longitudinal ligament, the yellow ligament, and the supra- and inter-spinous ligaments, was denoted as the OA-, OY-, and OSIL-index, respectively. We also evaluated the extent to which OPLL occupied the canal in the cervical spine, and we classified CNR [7] at the most compressed segment, as follows: Grade 1, 0% < CNR ≤ 25%; Grade 2, 25% < CNR ≤ 50%; Grade 3, 50% < CNR ≤ 75%; and Grade 4, 75% < CNR. We categorized diffuse idiopathic skeletal hyperostosis (DISH) according to the extension of ossified bridging as follows: Grade 1, DISH distributed at T3-10; Grade 2, DISH distributed at C6-T10 or T3-L2; and Grade 3, DISH distributed at C1-T10 or T3-S1 [14].

### 2.4. Statistical Analysis

The χ^2^ test was used for the statistical analysis of the difference in sex, DM, and prevalence of each pain. Kruskal–Wallis analysis was applied to compare three groups in the Cervical OP-index classification and four groups in the CNR grade using SPSS for Windows version 22.0 (SPSS Institute, Chicago, IL, USA). All data are expressed as the mean ± standard deviation (SD). A *p* value less than 0.05 was considered statistically significant.

## 3. Results

### 3.1. Demographic Data

In total, 239 patients were included in this study. Patient demographics are shown in Table 1 and Table 2. According to the cervical OP-index grading, 113 patients were Grade 1, 90 were Grade 2, and 36 were Grade 3. The mean age was 62.7 years for Grade 1, 64.9 years for Grade 2, and 65.3 years for Grade 3. The prevalence of DM was 18.6% in Grade 1, 31.1% in Grade 2, and 27.8% in Grade 3. BMI was similar among all three grades. The mean cervical JOA score was 12.4 points in Grade 1, 12.4 points in Grade 2, and 11.6 points in Grade 3. The prevalence of neck pain, back pain, and LBP was 65.5%, 25.7%, and 50.4% in Grade 1; 54.4%, 26.7%, and 60% in Grade 2; and 48.7%, 38.9%, and 47.2% in Grade 3, respectively. The mean OP-index of the thoracic, lumbar, and whole spine was 1.1, 0.1, and 4.7 in Grade 1; 2.6, 0.7, and 10.5 in Grade 2; and 4.3, 0.8, and 16.1 in Grade 3, respectively. The mean DISH grade was 0.4 for Grade 1, 1.0 for Grade 2, and 1.5 for Grade 3; the cervical OP-index was strongly correlated with each radiologic finding.

With regards to CNR grading (Table 2), 71 patients were Grade 1, 82 were Grade 2, 62 were Grade 3, and 17 were Grade 4. The mean age was 63.4 years for Grade 1, 65.4 years for Grade 2, 64.0 years for Grade 3, and 59.0 years for Grade 4. There were no significant differences in terms of comorbid DM, BMI, and cervical JOA scores among the four grades. The prevalence of neck pain, back pain, and LBP was 53.5%, 19.7%, and 50.7% in Grade 1; 59.8%, 34.1%, and 50% in Grade 2; 67.3%, 29.0%, and 59.7% in Grade 3; and 70.6%, 29.4%, and 70.6% in Grade 4, respectively. The mean OP-index of the cervical, thoracic, lumbar, and whole spine was 3.7, 1.1, 0.2, and 5.0 in Grade 1; 6.5, 2.5, 0.5, and 9.5 in Grade 2; 7.2, 2.8, 0.8, and 10.7 in Grade 3; and 7.5, 2.2, 0.6, and 10.6 in Grade 4, respectively. The mean DISH grade was 0.5 for Grade 1, 1.0 for Grade 2, 1.2 for Grade 3, and 0.5 for Grade 4. The CNR grade was significantly correlated with only the cervical OP-index in radiologic findings.

### 3.2. Cervical OP-Index Grade Was Not Associated with Body Pain in Patients with Cervical OPLL

To clarify associations between ossification predisposition and body pain, we investigated the VAS scores for neck pain, upper extremity numbness, and LBP. The mean VAS in Grades 1, 2, and 3 was 39.4, 38.4, and 37.4 for neck pain (Figure 2a); 44.8, 46.3, and 42.3 for upper extremity numbness (Figure 2b); and 22.5, 24.0, and 23.5 for LBP (Figure 2c), respectively. There was no significant correlation between the cervical OP-index grade and the pain score for each item.

### 3.3. Cervical OP-Index Grade Was Associated with Lower Extremity Function, Social Dysfunction, and Locomotive Function

JOA-CMEQ and JOA-BPEQ were used to evaluate ADL in OPLL patients and to investigate whether radiologic findings correlated with the restriction of ADL.

For JOA-CMEQ, there were no significant correlations except for lower extremity function (Figure 3a–e). Cervical function tended to deteriorate with an increasing cervical OP-index grade but with no significant difference. Notably, patients categorized as Grade 3 had significantly poorer cervical function (mean, 56.7, Figure 3a) than patients with Grades 1 (mean, 67.7) and 2 (mean, 68.1). The mean score of lower extremity function was 68.1 for Grade 1, 67.1 for Grade 2, and 55.2 for Grade 3 (Figure 3c).

For JOA-BPEQ (Figure 4a–e), there were no significant correlations in lumbar function, cognition, and body pain (Figure 4a,c,e). However, the mean score in Grades 1, 2, and 3 was 61.1, 51.0, and 49.8 for social dysfunction, and 68.6, 62.1, and 52.7 for locomotive function, respectively, showing a significant correlation between the cervical OP-index grade and these two items (Figure 4b,d).

### 3.4. CNR in the Cervical Spine Did Not Correlate with Clinical Symptoms

To clarify the clinical symptoms and OPLL thickness in the cervical spine, we investigated for a relationship between CNR and each item of the patient-reported evaluation. However, there was no significant correlation between the CNR grade and the pain score for each item (Figure 5a–c). Similarly, no statistical associations were found between the CNR grade and the patient-reported evaluation, including JOA-CMEQ (Figure 6a–e) and JOA-BPEQ (Figure 7a–e). A comparison between CNR Grade 4 and ≤Grade 3 revealed a significantly lower score in Grade 4 for lower extremity function of the JOA-CMEQ (Figure 6c) and locomotive function of the JOA-BPEQ (Figure 7d).

### 3.5. Cervical OP-Index Grade Correlates with Whole-Spine Radiologic Findings

The cervical OP-index grade was also closely related to OPLL in the thoracic and lumbar spine (Figure 8a,b). The CNR grade was 1.8 for Grade 1, 2.3 for Grade 2, and 2.8 for Grade 3 (Figure 8c). The cervical OP-index grade was significantly correlated with OPLL thickness. The DISH grade was also associated with the cervical OP-index grade (Figure 8d).

## 4. Discussion

Various studies have examined surgical techniques, [2] postoperative courses [6,7,21], perioperative complications [3], and types of ossification [22]; however, no study has investigated the association between radiologic severity of OPLL and clinical symptoms. To our knowledge, this is the first study to evaluate, with surveillance, this relationship based on multicenter large data in a prospective manner. We found that approximately half of the patients with a diagnosis of cervical OPLL had neck pain and/or LBP, regardless of the number of ossified lesions. Nakajima et al. [23] conducted a nationwide epidemiologic survey that included 3401 patients with spinal cord-related disorders and demonstrated that 49.5% of patients had some pain associated with the spinal lesion. Compared with the results of surveys in healthy volunteers [10,11], patients with cervical OPLL were more likely to develop body pain. Interestingly, patients categorized as Grade 3 on the cervical OP-index were less likely to have neck pain compared with patients classified as Grade 1 in this study. Fujimori et al. [24] compared a group of patients with OPLL and another with cervical spondylotic myelopathy (CSM), reporting higher VAS scores for neck pain in the CSM group. Another study [25] compared patients with OPLL and others with CSM using propensity score matching and demonstrated that CSM patients were more likely to complain of neck pain, numbness in all four limbs, and lower extremity pain compared with OPLL patients. This suggests that patients with preserved cervical spine mobility might have more neck pain compared with those with cervical spine immobility, such as those with a high cervical OP-index. In this series, we did not evaluate range of motion of the cervical spine for medical safety reasons in this series.

The findings of this surveillance study confirmed an association of the cervical OP-index with the thoracic and lumbar OP-index, which was in agreement with previous retrospective studies [12,17], and indicates that ossification predisposition is consistent in the whole spine. We also found that the cervical OP-index grade significantly correlated with the CNR grade. Park et al. [26] tracked 97 cervical OPLL patients (mean follow-up of 39.3 months) and observed a higher progression of ossified lesions in younger patients and those with C2-3 involvement. They also reported that ossification with a morphology crossed the segment but did not fuse with a segmental range of motion >5° nor with an initial thickness >5 mm, both of which were risk factors for chronological OPLL progression. These findings indicate increased thickness ossification in patients with more extensive OPLL in a cranio-caudal direction, referred to as a high OP-index. Additionally, the prevalence of DISH also correlates with the cervical OP-index grade. Nishimura et al. [14] reported that DISH prevalence was as high as 50% in cervical OPLL patients and the DISH grade was closely associated with the cervical OP-index grade. Mori et al. [15] reviewed data of an OPLL patient population and demonstrated a close association of the degree of ossification of the supra- and inter-spinous ligaments with that of cervical OPLL. Thus, ossified lesions in each spinal ligament correlated with each other.

This study demonstrated that the degree of body pain based on a VAS score was not associated with the OP-index or CNR in OPLL patients. Similarly, there was no significant correlation between upper extremity function on the JOA-CMEQ and cervical OP-index grade or CNR grade. However, with regards to the prevalence of pain symptoms, neck pain and LBP tended to be more frequent in patients with a higher CNR grade, although this was not statistically significant. We did not find this tendency in the cervical OP-index grade. These results indicate the following two points: the bias that most OPLL patients in this study had neck pain and/or upper extremity numbness, which might have masked associations between the severity of these symptoms and the radiologic severity of ossification; and pain symptoms in patients with OPLL that might be influenced by the thickness of ossified lesions compressing the spinal cord rather than extending ossified lesions. In addition, our findings demonstrated that the CNR grade appeared to negatively correlate with the JOA score, which reflected neurologic status. Various studies have reported that massive OPLL can compress the spinal cord and induce myelopathic symptoms. Therefore, it is vital to pay attention to patients with a high CNR grade in order to prevent neurologic deterioration.

In this survey, the cervical OP-index grade significantly correlated with lower extremity function, social dysfunction, and locomotive function. Similarly, our findings revealed that patients categorized as Grade 4 CNR were likely to deteriorate in terms of lower extremity function and locomotive function. Ito et al. [27] reviewed CT myelography in 41 cervical OPLL patients with a weak positive correlation between the JOA score and the cross-sectional area of the cord at the narrowest segment. Kameyama et al. [28] analyzed nine autopsy OPLL cases to elucidate the relationship between spinal cord morphology and pathology. Major pathological changes were restricted to the grey matter and were absent or minimal in the white matter in cases of spinal cord compression by only an anterior ossified lesion. More severe pathologic findings were seen in the lateral pyramidal tracts and the posterior column in cases with spinal cord compression in the anterior portion as well as the lateral surfaces. Furthermore, they demonstrated that the spinal cord compression ratio (sagittal diameter/transverse diameter × 100) was unrelated to pathological changes. These previous data and our results suggest that the degree of spinal disorder may be influenced not only by the size of the ossified lesions but also by the lateral development of OPLL.

This study had several limitations. First, it was not population-based and was conducted prospectively. Second, it was not a longitudinal study. Third, we only investigated the severity of symptoms but not detailed characteristics of symptoms in OPLL patients. Therefore, there could be pathological heterogeneity in the patients overall. Fourth, we could not evaluate ossified lesions on the axial CT imaging to clarify whether OPLL develops laterally. In addition, we could not determine whether the mobility of the segment affected by OPLL affects the pain and severity of myelopathy. Nevertheless, we believe that our findings provide important information concerning the diagnostic features of patients with OPLL.

## Figures and Tables

**Figure 1 jcm-09-04055-f001:**
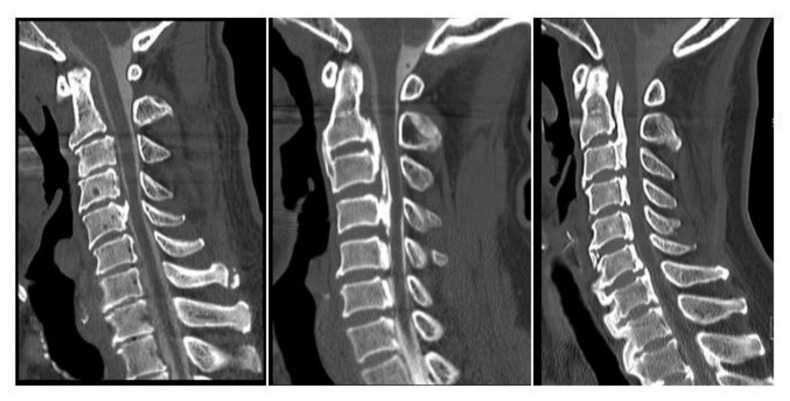
The cervical OPLL ossification index (OP-index) classification.

**Figure 2 jcm-09-04055-f002:**
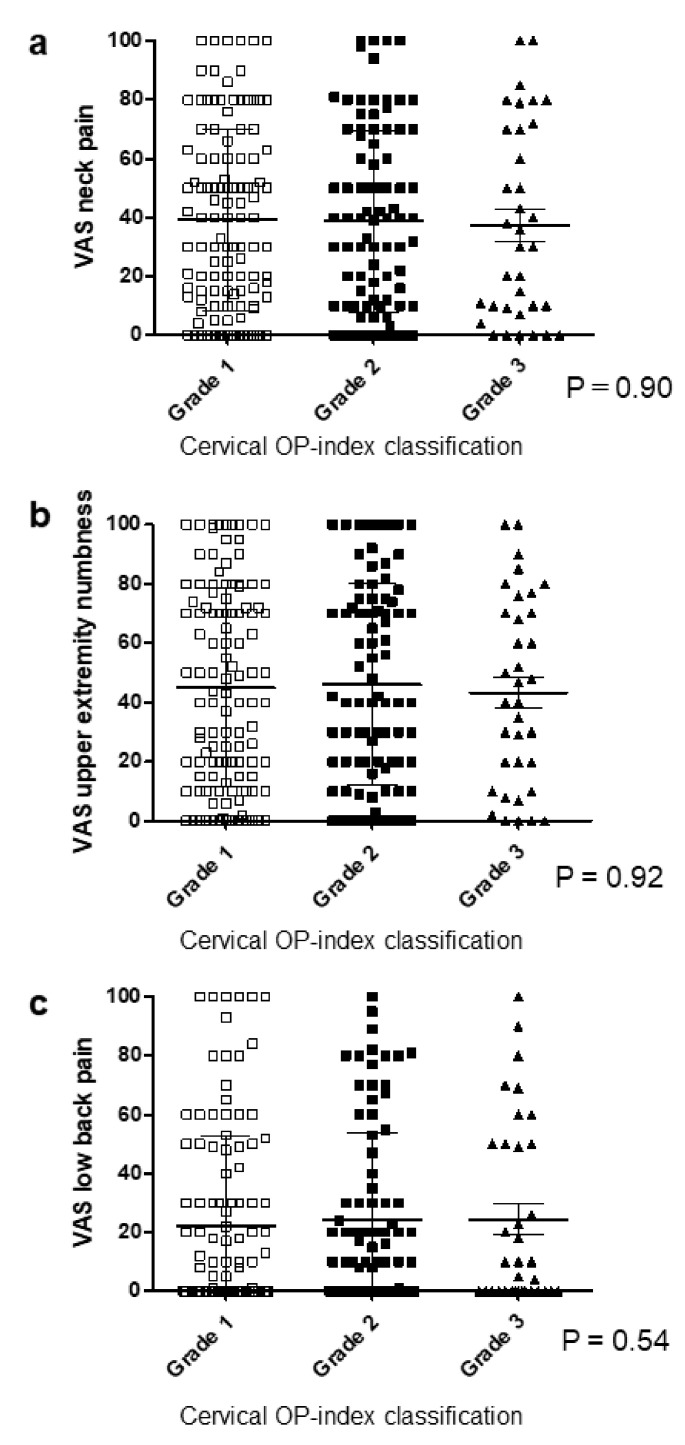
Relationship between the VAS score and the cervical OP-index grade. (**a**) VAS neck pain; (**b**) VAS upper extremity numbness; (**c**) VAS low back pain. OP-index, ossification index of OPLL; VAS, visual analog scale.

**Figure 3 jcm-09-04055-f003:**
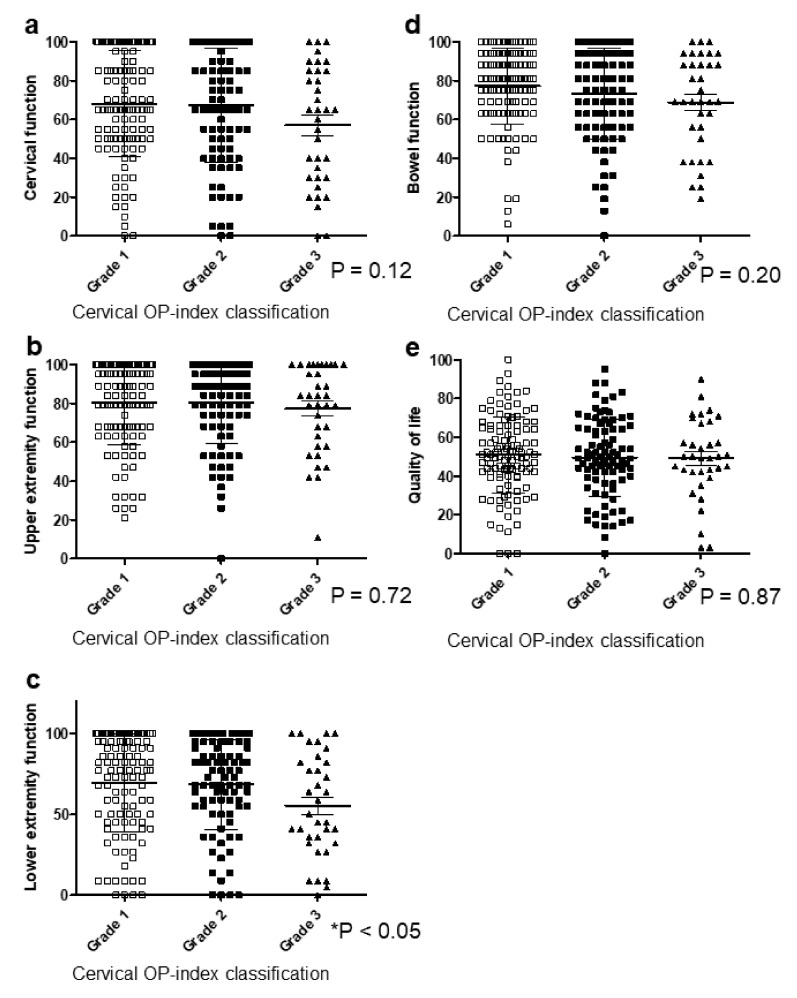
Relationship between JOA-CMEQ and the cervical OP-index grade. (**a**) Cervical function; (**b**) upper extremity function; (**c**) lower extremity function; (**d**) bowel function; (**e**) quality of life. JOA-CMEQ, Japanese Orthopedic Association Cervical Myelopathy Evaluation Questionnaire; OP-index, ossification index of OPLL.

**Figure 4 jcm-09-04055-f004:**
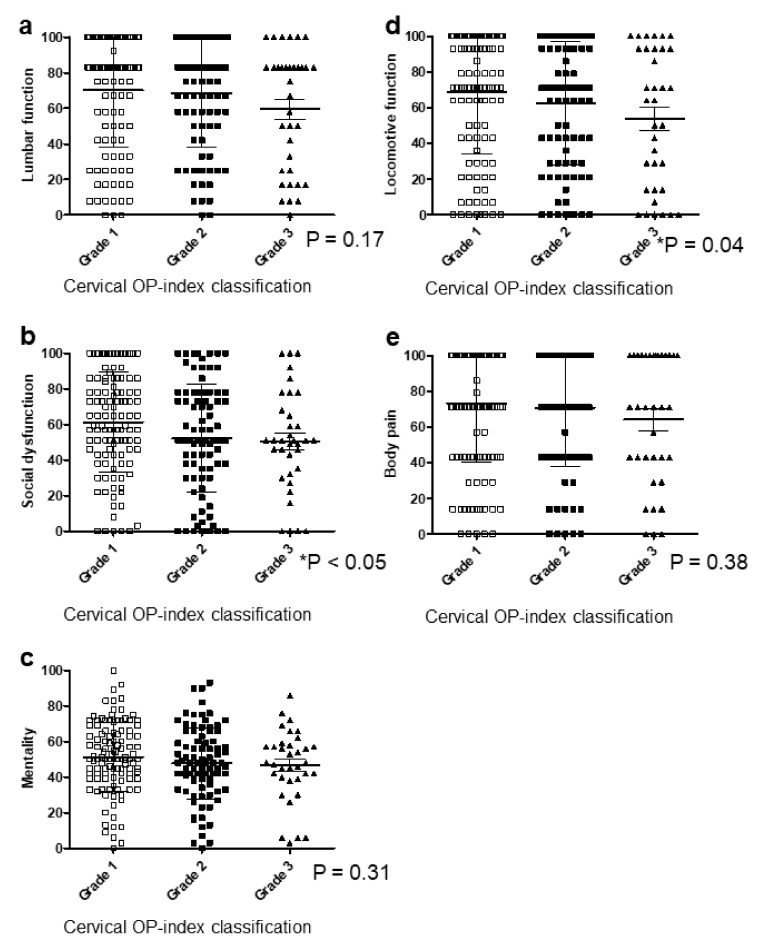
Relationship between JOA-BPEQ and the cervical OP-index grade. (**a**) Lumbar function; (**b**) social dysfunction; (**c**) mentality; (**d**) locomotive function; (**e**) body pain. JOA-BPEQ, Japanese Orthopedic Association Back Pain Evaluation Questionnaire; OP-index, ossification index of OPLL.

**Figure 5 jcm-09-04055-f005:**
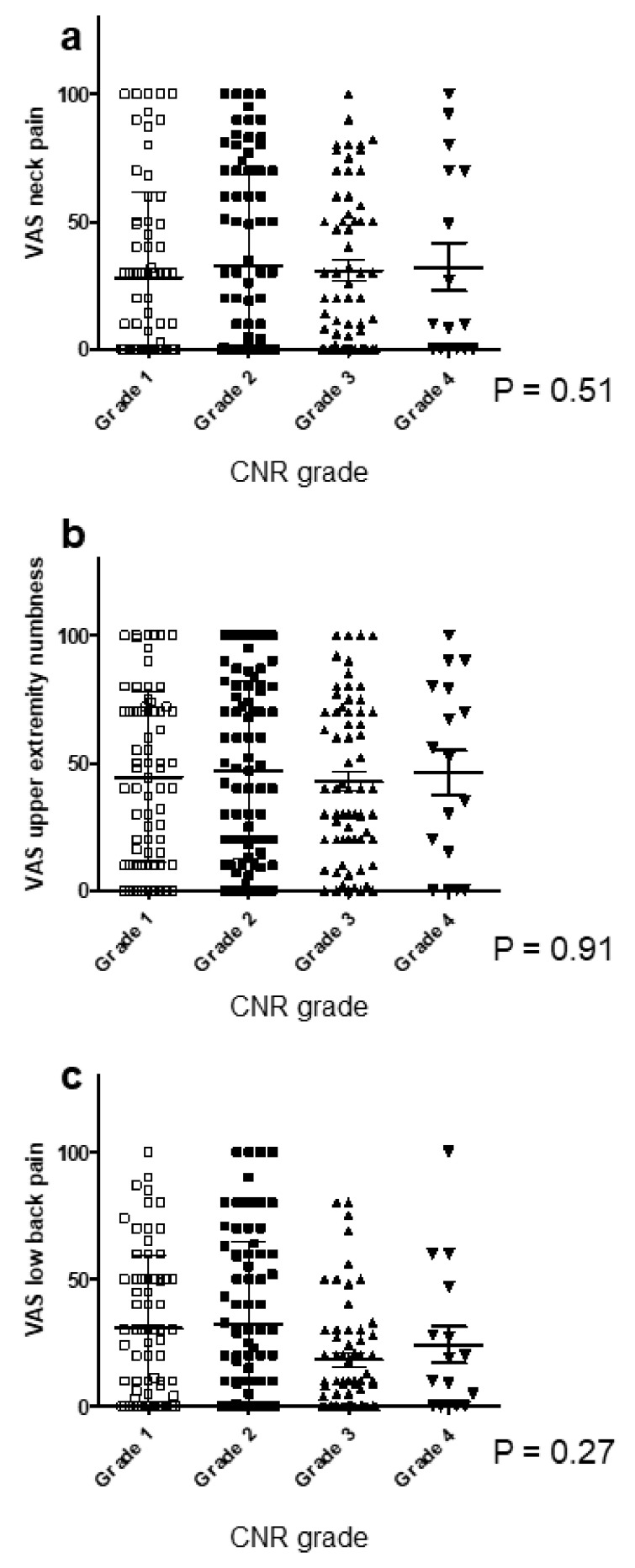
Relationship between the VAS score and the CNR grade. (**a**) VAS neck pain; (**b**) VAS upper extremity numbness; (**c**) VAS low back pain. CNR, canal narrowing ratio; VAS, visual analog scale.

**Figure 6 jcm-09-04055-f006:**
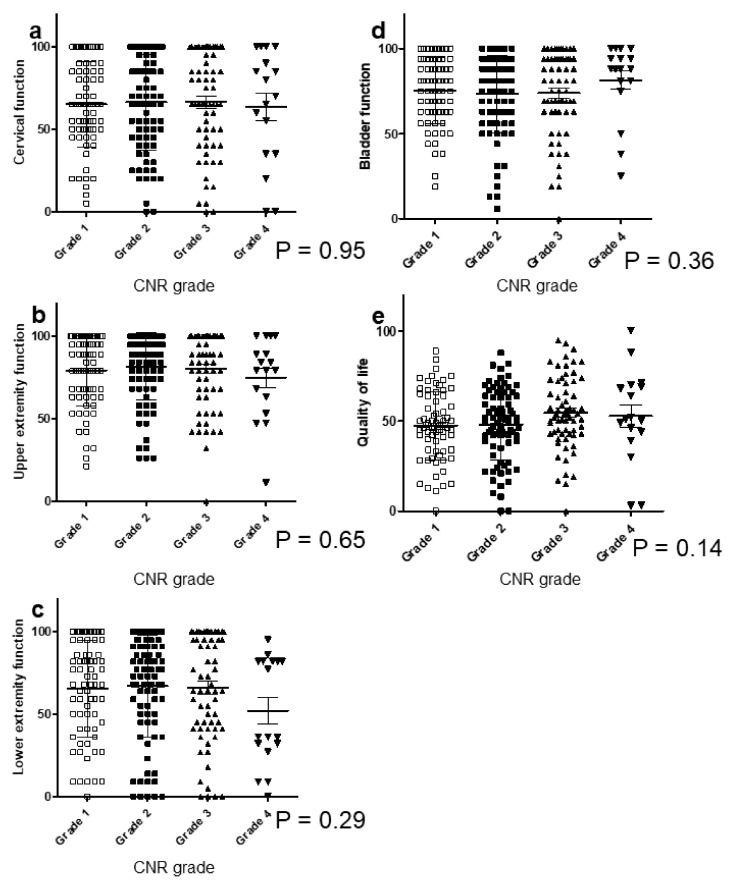
Relationship between JOA-CMEQ and the CNR grade. (**a**) Cervical function; (**b**) upper extremity function; (**c**) lower extremity function; (**d**) bladder function; (**e**) quality of life. CNR, canal narrowing ratio; JOA-CMEQ, Japanese Orthopedic Association Cervical Myelopathy Evaluation Questionnaire.

**Figure 7 jcm-09-04055-f007:**
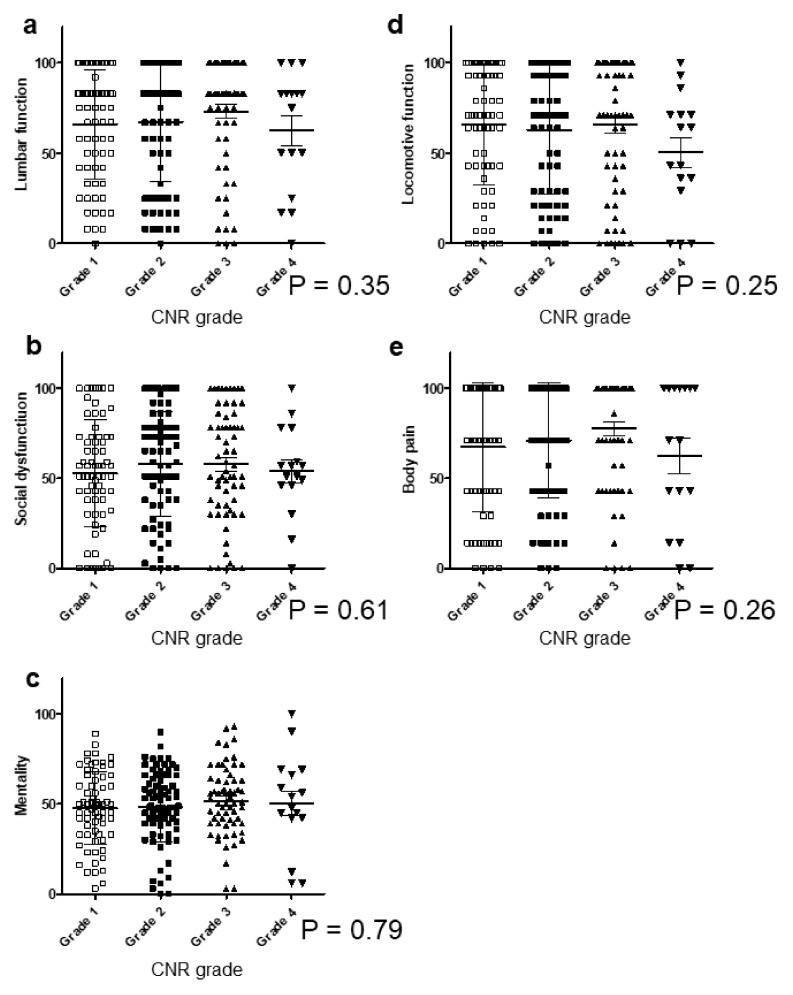
Relationship between JOA-BPEQ and the CNR grade. (**a**) Lumbar function; (**b**) social dysfunction; (**c**) mentality; (**d**) locomotive function; (**e**) body pain. CNR, canal narrowing ratio; JOA-BPEQ, Japanese Orthopedic Association Back Pain Evaluation Questionnaire.

**Figure 8 jcm-09-04055-f008:**
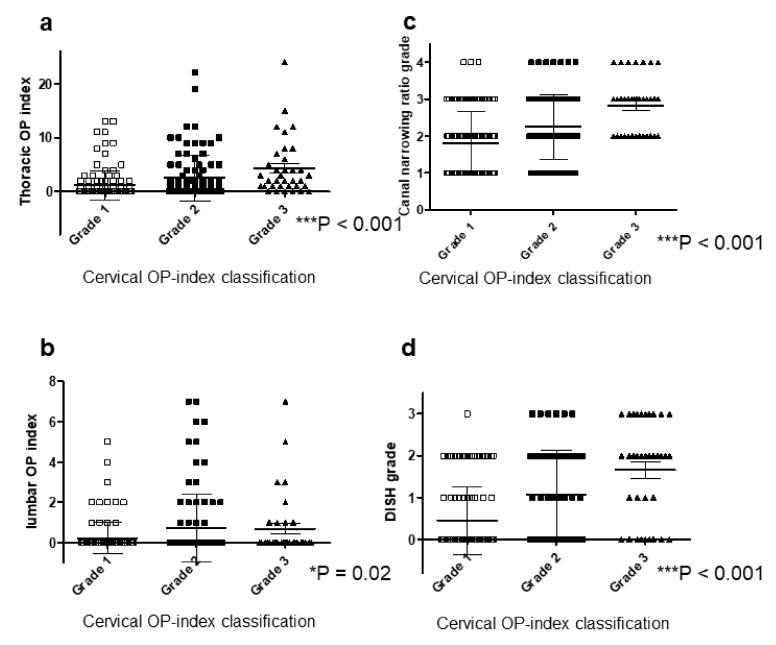
Relationship between the cervical OP-index grade and other radiologic findings. (**a**) Thoracic OP-index; (**b**) lumbar OP-index; (**c**) CNR grade; (**d**) DISH grade. CNR, canal narrowing ratio; DISH, diffuse idiopathic skeletal hyperostosis; OP-index, ossification index of OPLL.

**Table 1 jcm-09-04055-t001:** Demographics of patients with cervical ossification of the posterior longitudinal ligament (OPLL) according to the cervical OP-index grading system.

	Grade 1 (*n* = 113)	Grade 2 (*n* = 90)	Grade 3 (*n* = 36)	*p*
Age (years)	62.7 ± 11.9	64.9 ± 12.3	65.3 ± 13.4	0.88
Sex (M/F)	75/38	65/25	23/13	0.08
Diabetes mellitus (%)	18.6%	31.1%	27.8%	0.12
BMI	25.3 ± 4.2	26.5 ± 4.5	26.4 ± 5.3	0.74
Cervical JOA score	12.4 ± 3.5	12.4 ± 3.2	11.6 ± 3.6	0.21
Prevalence of symptoms				
Neck pain	65.5%	54.4%	48.7%	0.15
Back pain	25.7%	26.7%	38.9%	0.13
Low back pain	50.4%	60%	47.2%	0.21
OP-index				
Thoracic spine	1.1 ± 2.8	2.6 ± 4.2	4.3 ± 0.8	<0.001
Lumbar spine	0.1 ± 0.9	0.7 ± 1.7	0.8 ± 1.5	0.02
Whole spine	4.7 ± 3.6	10.5 ± 5.7	16.1 ± 1.2	<0.001
DISH grade	0.4 ± 0.8	1.0 ± 1.1	1.5 ± 1.2	<0.001

Data are expressed as the mean ± standard deviation; BMI, body mass index; DISH, diffuse idiopathic skeletal hyperostosis; JOA, Japanese Orthopedic Association; OP-index, ossification index of OPLL.

**Table 2 jcm-09-04055-t002:** Demographics of patients with cervical OPLL according to the CNR grading system.

	Grade 1 (*n* = 71)	Grade 2 (*n* = 82)	Grade 3 (*n* = 62)	Grade 4 (*n* = 17)	*p*
Age (years)	63.4 ± 12.6	65.4 ± 10.8	64.0 ± 13.6	59.0 ± 11.9	0.14
Male (M/F)	45/26	56/26	46/16	13/4	0.10
Diabetes mellitus (%)	16.9%	31.7%	24.2%	35.3%	0.17
BMI	25.5 ± 4.6	26.3 ± 4.5	25.4 ± 4.2	27.4 ± 4.8	0.11
Cervical JOA score	12.3 ± 3.0	12.4 ± 3.2	11.6 ± 3.6	11.9 ± 3.3	0.07
Prevalence of symptoms					
Neck pain	53.5%	59.8%	67.7%	70.6%	0.14
Back pain	19.7%	34.1%	29.0%	29.4%	0.11
Low back pain	50.7%	50%	59.7%	70.6%	0.17
OP-index					
Cervical spine	3.7 ± 2.3	6.5 ± 3.0	7.2 ± 3.0	7.5 ± 3.0	0.02
Thoracic spine	1.1 ± 2.6	2.5 ± 4.5	2.8 ± 4.4	2.2 ± 3.2	0.24
Lumbar spine	0.2 ± 0.8	0.5 ± 1.2	0.8 ± 1.7	0.6 ± 1.6	0.08
Whole spine	5.0 ± 3.9	9.5 ± 6.6	10.7 ± 6.7	10.6 ± 6.1	0.07
DISH grade	0.3 ± 0.8	1.0 ± 1.1	1.2 ± 1.1	0.5 ± 0.8	0.19

Data are expressed as the mean ± standard deviation; OP-index, ossification index of OPLL; BMI, body mass index; DISH, diffuse idiopathic skeletal hyperostosis; JOA, Japanese Orthopedic Association; CNR, canal narrowing ratio.

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
