# Peer review of "Associations between Clinical Symptoms and Degree of Ossification in Patients with Cervical Ossification of the Posterior Longitudinal Ligament: A Prospective Multi-Institutional Cross-Sectional Study"

_jcm, 2020, doi:10.3390/jcm9124055_

Round 1

Reviewer 1 Report

This article is a multicenter, prospective, observational study.

Authors examines the relationship between radiological characteristics and clinical symptoms, such as pain and physical function, in patients with OPLL who did not lead to surgery. Results showed that the degree of ossification (OPindex) was not associated with pain, and correlated with lower extremity functional decline and social dysfunction. In addition, spinal canal width was associated with lower extremity functional decline.

Overall, it is a well-designed study, but there are problems.

1, One of the problems is the exclusion of patients who had undergone or were undergoing surgery: if the degree of OPLL progression and spinal occupancy were involved in clinical symptoms, there would have been more clinical changes in patients who were indicated for surgery. Therefore, the target population in this study was patients with mild disease without an indication for surgery. At least, the authors should state a clear indication for surgery. In addition, the title or abstract should include the meaning "in patients with OPLL who did not require surgery".

Second, association between cervical mobility and pain or clinical symptoms has not been shown. The problem with ligamentous ossification is the loss of mobility at the ossification site and compression on the spinal cord and increased mechanical stress on adjacent vertebrae outside the ossification site. Since the pain is not separated into resting pain and motion pain, it is difficult to know whether the pain is due to pressure on the spinal cord or nerve roots from the ossification or from adjacent vertebrae outside of the ossified area, such as facet pain or discogenic pain. If you want to examine the relationship between the development of ligamentous ossification, narrowing of the spinal canal and pain and spinal cord function, it seems imperative to consider them.

There are numerous factors that contribute to neck and back pain, and the degree of ossification of OPLL is only one of them. This study only assumes that it was negative and is likely to have little clinical significance. It has also been reported in the past that a higher degree of ossification leads to a decrease in lower limb function, so this is not new information.

Author Response

Reviewer #1

This article is a multicenter, prospective, observational study.

Authors examines the relationship between radiological characteristics and clinical symptoms, such as pain and physical function, in patients with OPLL who did not lead to surgery. Results showed that the degree of ossification (OPindex) was not associated with pain, and correlated with lower extremity functional decline and social dysfunction. In addition, spinal canal width was associated with lower extremity functional decline.

Overall, it is a well-designed study, but there are problems.

・1, One of the problems is the exclusion of patients who had undergone or were undergoing surgery: if the degree of OPLL progression and spinal occupancy were involved in clinical symptoms, there would have been more clinical changes in patients who were indicated for surgery. Therefore, the target population in this study was patients with mild disease without an indication for surgery. At least, the authors should state a clear indication for surgery. In addition, the title or abstract should include the meaning "in patients with OPLL who did not require surgery".

Response:

We thank Reviewer#1 for taking time to review our paper. This study basically included OPLL patients regardless of whether operation was required or not. To make readers understand easily, we put the sentence ‘regardless of whether operation was required or not’ in page 3 lines116-117.

・Second, association between cervical mobility and pain or clinical symptoms has not been shown. The problem with ligamentous ossification is the loss of mobility at the ossification site and compression on the spinal cord and increased mechanical stress on adjacent vertebrae outside the ossification site. Since the pain is not separated into resting pain and motion pain, it is difficult to know whether the pain is due to pressure on the spinal cord or nerve roots from the ossification or from adjacent vertebrae outside of the ossified area, such as facet pain or discogenic pain. If you want to examine the relationship between the development of ligamentous ossification, narrowing of the spinal canal and pain and spinal cord function, it seems imperative to consider them.

Response:

Thank you for giving us a helpful comment and good insight. As the Reviewer #2 pointed out, mobility in the segments affected is a significant factor of development and deterioration of myelopathy. We actually did not investigate detailed data regarding mobility of cervical spine. This study covers patients ranging from mild to very severe symptoms. According to medical safety reason, therefore, the study did not include functional X-ray. In previous study, we compared OPLL patients and cervical spondylosis patients, non OPLL where mobility of the cervical spine is preserved, under propensity matching cohort analysis and demonstrated that neck paint and upper extremity numbness were worse in cervical spondylosis cases. Therefore, we have stated the sentence ‘This suggests that patients with preserved cervical spine mobility might have more neck pain compared with those with cervical spine immobility, such as those with high cervical OP-index, although we did not evaluate range of motion of the cervical spine in this series.’ in page 12, line 274. Although this evidence cannot directly explain that mobility in the segment, it might provide a speculation that mobility of cervical spine at the segment affected is one of important factors for deterioration of pain and myelopathic symptoms. In addition, we added ‘for medical safety reasons’ in page 12 line 274, and ‘In addition, we could not identify whether segmental mobility of OPLL affects pain and serenity of myelopathy’ in page 13 lines 322-323 to mention the limitation in this study.

・There are numerous factors that contribute to neck and back pain, and the degree of ossification of OPLL is only one of them. This study only assumes that it was negative and is likely to have little clinical significance. It has also been reported in the past that a higher degree of ossification leads to a decrease in lower limb function, so this is not new information.

Response:

As the Reviewer #1 described, the results of our study demonstrated little significance on clinical symptoms in OPLL patients. Although various investigations have shown that the degree of ossification correlates with severity of neurologic symptoms, we believe that there has been no multi-center study including both radiologic and patient-reported clinical outcomes OPLL patients. We would like to emphasize this point in this paper.

Reviewer 2 Report

 The purpose of this study was to examine the associations between clinical symptoms and degree of ossification in cervical OPLL patients. 239 patients were evaluated.

  1. Dynamic factors have been previously shown to relate to the development of myelopathy in cervical OPLL patients. What percentage of the patients in each grade of CNR classification had motion segment at the peak of the ossification? Are there any differences of JOA-CMEQ and JOA-BPEQ results between the "motion segment" group and "fusion" group in the same CNR grade?

  1. The results showed that there were no significant correlations between upper extremity function in JOA-CMEQ and cervical OP-index grade or CNR grade. These results may mislead the readers that the degree of ossification does not relate to upper extremity function. Please comment.

Author Response

The purpose of this study was to examine the associations between clinical symptoms and degree of ossification in cervical OPLL patients. 239 patients were evaluated.

・Dynamic factors have been previously shown to relate to the development of myelopathy in cervical OPLL patients. What percentage of the patients in each grade of CNR classification had motion segment at the peak of the ossification? Are there any differences of JOA-CMEQ and JOA-BPEQ results between the "motion segment" group and "fusion" group in the same CNR grade?

Response:

We greatly thank Reviewer #2 for taking time to review our paper. As the Reviewer #2 pointed out, mobility in the segments affected is a significant factor of development and deterioration of myelopathy. We actually did not investigate detailed data regarding mobility of cervical spine. This study covers patients ranging from mild to very severe symptoms. According to medical safety reason, therefore, the study did not include functional X-ray. In previous study, we compared OPLL patients and cervical spondylosis patients, non OPLL where mobility of the cervical spine is preserved, under propensity matching cohort analysis and demonstrated that neck paint and upper extremity numbness were worse in cervical spondylosis cases (page 12, line 274). Although this evidence cannot directly explain that mobility in the segment, it might provide a speculation that mobility of cervical spine at the segment affected is one of important factors for deterioration of pain and myelopathic symptoms. In addition, we added ‘for medical safety reasons’ in page 12 line 274, and ‘In addition, we could not identify whether segmental mobility of OPLL affects pain and serenity of myelopathy’ in page 13 lines 322-323 to mention the limitation in this study.

・The results showed that there were no significant correlations between upper extremity function in JOA-CMEQ and cervical OP-index grade or CNR grade. These results may mislead the readers that the degree of ossification does not relate to upper extremity function. Please comment.

Response:

As the Reviewer suggested, we added the sentence ‘a biased fact that most OPLL patients in this study had neck pain and/or upper extremity numbness might masks associations between the degrees of these symptoms and radiologic severity of ossification’ in page 13, lines 295-299. We also added ‘Similarly, there were no significant correlations between upper extremity in JOA-CMEQ and cervical OP-index grade or CNR grade.’ in page 13 lines 291-292.  

Round 2

Reviewer 1 Report

In reply, I understood the authors' intentions. I don't think the revised version has any major problems.

Reviewer 2 Report

The authors clarify my concerns.